# Devising Mapping Interoperability with Mapping Translation

Ana Iglesias-Molina*1*,  Andrea Cimmino*1* and  Oscar Corcho*1*

*1Ontology Engineering Group, Universidad Politécnica de Madrid*

## Abstract
Nowadays, Knowledge Graphs are extensively created using very different techniques, mapping languages among them. The wide variety of use cases, data peculiarities, and potential uses has had a substantial impact in how these languages have been created, extended, and applied. This situation is closely related to the global adoption of these languages and their associated tools. The large number of languages, compliant tools, and usually the lack of information of the combination of both leads users to use other techniques to construct Knowledge Graphs. Often, users choose to create their own ad hoc programming scripts that suit their needs. This choice is normally less reproducible and maintainable, what ultimately affects the quality of the generated RDF data, particularly in long-term scenarios. We devise with mapping translation an enhancement to the interoperability of existing mapping languages. This position paper analyses the possible language translation approaches, presents the scenarios in which it is being applied and discusses how it can be implemented.

## Keywords
Mapping languages, Ontology Description, Mapping Translation

## 1. Introduction

Knowledge Graphs (KG) are increasingly used in academia and industry to represent and manage the increasing amount of data on the Web [1]. A large number of techniques to create KGs have been proposed. These techniques may follow, namely, two approaches: RDF materialization, that consists of translating data from one or more heterogeneous sources into RDF; or Virtualization, (Ontology Based Data Access) [2] that consists in translating a SPARQL query into one or more equivalent queries which are distributed and executed on the original data source(s) and where its results are transformed back to the SPARQL results format [3]. Both approaches rely on an essential element, a mapping document, which is the key-enabler for performing the translations.

Mapping languages represent the relationships between the structure or the model of heterogeneous data and an RDF version following an ontology, i.e., the rules on how to translate from non-RDF data into RDF. This data can be originally expressed in a variety of formats, such as tabular, JSON, or XML. Due to the heterogeneous nature of data, the wide corpus of

*Third International Workshop On Knowledge Graph Construction, Co-located with the ESWC 2022, Crete - 30th May 2022*

✉ ana.iglesiasm@upm.es (A. Iglesias-Molina); andreajesus.cimmino@upm.es (A. Cimmino); oscar.corcho@upm.es (O. Corcho)

🆔 0000-0001-5375-8024 (A. Iglesias-Molina); 0000-0002-1823-4484 (A. Cimmino); 0000-0002-9260-0753 (O. Corcho)

techniques and the specific requirements that some scenarios may impose, an increasing number of mapping languages have been proposed [4, 5]. The differences among them are usually based on three aspects: (a) the focus on one or more particular data formats, e.g., the W3C Recommendations R2RML focuses on SQL tabular data [6]; (b) an addressed specific feature, e.g. SPARQL-Generate [7] allows the definition of functions in the mapping for cleaning or linking the generated RDF data; or (c) if they are designed for a particular technique or scenario that has special requirements, e.g. the WoT-mappings [8] where designed as an extension of the WoT standard [9].

As a result, the diversity of mapping languages allows the construction of KG from heterogeneous data sources in many different scenarios. Current mapping languages may be categorized by their schema: RDF-based (e.g. R2RML [6] and extensions, CSVW [10]), SPARQL-based (e.g., SPARQL-Generate [7], SPARQL-Anything [11]) or based on other schemas (e.g. ShExML [12], Helio mappingsHelio[1]). Nevertheless, the existing techniques usually implement just one mapping language, and sometimes not even the whole language specification [13]. Deciding which language and technique should be used in each scenario becomes a costly task, since the choice of one language may not cover all needed requirements [14]. Some scenarios require a combination of mapping languages because of their differential features, which entails using different techniques. In many cases, this diversity leads to ad hoc solutions that reduce reproducibility, maintainability, and reusability [15].

The increasing and heterogeneous emergence of new use cases still motivates the community to keep developing solutions that are, more commonly than desired, not compatible with existing ones. This position paper develops the concept of mapping translation, proposed by Corcho et al. [16], a concept that can enhance the interoperability among existing mapping languages and thus, improve the user experience of these technologies by allowing communication and understanding among them. This paper presents some approaches for language translation, shows the current situations in which mapping translation is being applied and their benefits, and proposes different techniques to extend it to more languages.

The remaining of this article is structured as follows: Section 2 provides some insights about language translation and the situations in which it is being applied. Section 3 proposes three different techniques to address mapping translation at a larger scale. Finally, Section 4 draw some conclusions of the concepts presented in the paper.

## 2. Mapping translation: Context

In this section, we introduce mapping translation describing some approaches to language translation and present a set of scenarios in which mapping translation has been applied. Authors assume the reader is familiar with current mapping languages and their general characteristics.

---

[1]https://github.com/oeg-upm/helio/wiki/Streamlined-use-cases#materialising-rdf-from-csv--xml-and-json-files-using-rml

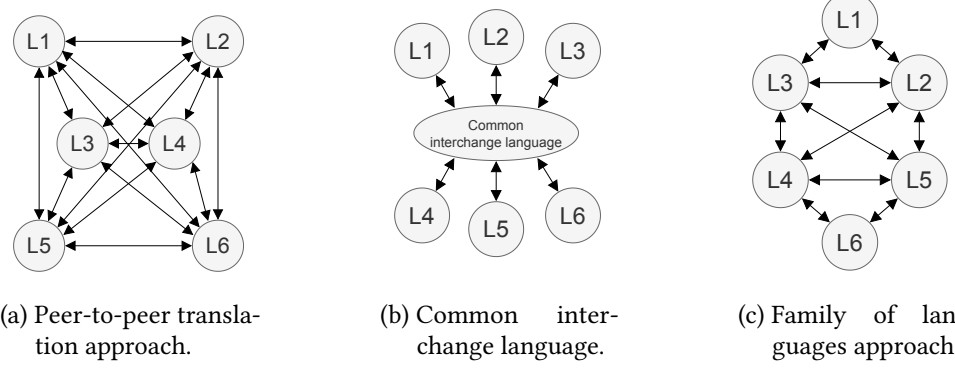

(a) Peer-to-peer transla-
tion approach.

(b) Common inter-
change language.

(c) Family of lan-
guages approach.

**Figure 1:** Types of language translations (Adapted from [18]).

## 2.1. Approaches to language translation

In the context of language translation, there are several approaches that carry out translations among a set of languages. Depending on the situation at hand, an approach can be advantageous with respect to the other ones. We highlight the following [17]:

**Peer-to-peer translation** (Fig. 1a) supports ad hoc translation solutions between pairs of languages. This one may seem as the most straightforward approach, requiring the development of only the translator services needed for the situation at hand and with the possibility of adjusting it ad hoc for each situation. However, it becomes decreasingly feasible as the number of required translations increases.

**Common interchange language** (Fig. 1b) uses a language that serves as an intermediary among several languages. This approach reduces the number of translator services needed to develop and it is the most feasible of the three to scale in amount. It involves creating (or luckily having) a language able to represent the expressiveness of all languages, to avoid information loss. Additionally, this implies that there are common patterns shared by the languages independently of their representation, and that an abstract manner of gathering them is possible, which may not be thus for highly heterogeneous languages.

**Family of languages** (Fig. 1c) considers sets of languages and translations between the representatives of each set. This approach stands out for situations where there are clear subgroups of languages similar among them but among languages from other groups.

## 2.2. Mapping translation scenarios

Regarding mapping languages, there are currently some implementations that unidirectionally translate pairs of mapping languages. ShExML and YARRRML in their respective online editors[2,3] enable translation to RML. Another case is when tools implement RML/R2RML mapping

---

[2]http://shexml.herminiogarcia.com/editor/
[3]https://rml.io/yarrrml/matey/#

translation into the language they are designed to parse; such is the case of Helio[4] and SPARQL-Generate[5], that translate from RML to their respective language; and Ontop [19], that translates R2RML into its proprietary language, OBDA mappings [20]. These translation makes it possible to extend the outreach of the tool, since they enable the possibility of using them without the need of learning their specific language, but using one that is widely used and extended, such as R2RML and RML.

Another case we want to present is Mapeathor [21], a tool that takes the mapping rules specified in spreadsheets and transforms them into a mapping in either R2RML, RML or YARRRML. It aims to lower the learning curve of those languages for new users and ease the mapping writing process. Finally, we remark the case where tools provide a set of optimizations on the construction of RDF graphs exploiting the translation of mapping rules, this is the case of Morph-CSV [22] and FunMap [23]. Morph-CSV first performs a transformation over the tabular data with RML+FnO mappings and CSVW annotations, and outputs a database and R2RML mappings ready to be transformed by an R2RML-compliant tool. FunMap takes an RML+FnO mapping, performs the transformation functions indicated, outputs the parsed data and generates a function-free RML mapping.

The approaches presented are, mainly, examples of peer-to-peer translation for specific uses. The exception is Mapeathor, that abstracts the rules from R2RML, RML and YARRRML in a spreadsheet-based representation, which aligns with the approach of a common interchange language. Even though most of these translation examples involve R2RML or RML, there is no holistic approach of a general translation framework.

## 3. Mapping translation: Techniques

This section presents three proposals to implement a mapping translator service general enough to enable translation among several languages. These proposals are, namely, (1) Software-based, (2) construct query-based, and (3) Executable mapping-based. These implementations can be applied to any of the language translation approaches presented in Section 2.1.

**Software-based translation.** It consists on ad-hoc software implementation for each pair of languages to perform bidirectional translations between them. As any ad hoc solution, it benefits from adjusting specifically to any situation with the (almost) unlimited possibilities that programming languages provide. This is the approach that all situations presented in Section 2.2 have applied, although with unidirectional translations.

**Construct query-based translation.** This approach takes advantage of SPARQL query language with construct queries, which return an RDF graph. These particular queries extract the data by matching graph patterns of the query (with the *WHERE* clause) and builds the output graph based on a template (with the *CONSTRUCT* clause). Since many languages are RDF-based, that is, follow the schema of an ontology and are usually written in the Turtle syntax (e.g., R2RML and extensions), this approach can be applicable to them. This approach benefits from relying on a well-stablished standard, as SPARQL is nowadays, and its compliant

---

[4]https://github.com/oeg-upm/helio/wiki/Streamlined-use-cases#materialising-rdf-from-csv--xml-and-json-files-using-rml

[5]https://github.com/sparql-generate/rml-to-sparql-generate

engines. However, it would leave out languages with other schemas, such as ShExML and SPARQL-based, wthout relying on software-based solutions.

**Executable mapping-based translation.** This last approach makes use of executable mappings automatically generated from ontology alignment to perform data translation between the two ontologies [24]. Similarly to the previous approach, this one also makes use of construct queries from SPARQL in the executable mappings. While the previous one relied on manual effort to build queries, this one takes advantage of the ontologies that define RDF-based mapping languages. In addition to the benefits and setbacks that the previous approach has, this approach may be hindered by the language constructs to build mappings. That is to say, single one-to-one correspondences of ontology entities may not be enough to gather and be able to translate their expressiveness and capabilities, especially for considerably different languages.

The techniques proposed are presented in decreasing order of manual effort required. The first one is completely ad hoc, and even though it could use some modules of the developed solutions presented in Section 2.2, many more would be needed to provide a complete set of bidirectional translations covering a good number of languages. The second one requires considerable effort to build queries for RDF-based languages, assuming no extra help from software implementation is needed. The third one could ideally be automatically done, from ontology alignments creation to mapping execution generation. However, the rate of success of this approach without manual intervention is not expected to be high, especially for the ontology alignment part when the input ontologies considerably differ from one another or present different constructs (with different number of elements or differently structured).

## 4. Conclusions

This paper develops the concept of mapping translation, proposed by Corcho et al. [16]. It analyses the possible language translation approaches, updates the scenarios in which it is being applied, and proposes some implementation techniques to perform it.

There are several possibilities in order to fully develop a complete solution to achieve mapping translation that ensures information preservation, as described in previous sections. It not only requires choosing the technical implementation according to the available efforts and resources, but more importantly, it involves deciding wisely the language translation approach that suits best this particular case of mapping languages. As presented previously, we categorize current mapping languages by their schema: RDF-based, SPARQL-based and based on other schemas. All of them have been designed for a basic purpose: describing non-RDF data to allow either materialization or virtualization. Intuitively, we can assume that the rules that the different mappings create can be represented in an abstract, language-independent manner. However, the sometimes large differences among these languages may question this assumption. Some languages, inside their categories, are similar to each other, R2RML and its extensions, for instance. Languages from different groups can be related, such as ShExML and RML, despite some inevitable differences in their features. There are others that are more unique, such as CSVW. Lastly, the SPARQL-based group is more isolated from the others due to the great possibilities that provide relying on SPARQL. This scenario poses challenges for every language translation approach. Peer-to-peer translation would require a substantial amount of effort

for divergent languages. Using families of languages would improve in comparison with the previous one, but it still would have to face several challenges in language representation and the amount of translator services required. Meanwhile, using a common interchange language would be the one that reduces most efforts, but there is no absolute certainty that a common interchange language could be able to represent them all. Still, some steps have been taken to draft this language[6], with the base idea that the mapping rules can be abstracted and represented in an ontology-based language.

Even though it does not present as an easy task, mapping translation is a concept that can only benefit the current landscape of heterogeneous mapping languages. After years of KG construction, in which the increasing and heterogeneous emergence of new use cases still motivates the community to keep developing solutions, sometimes ad hoc, sometimes with extensions of standards or widely used languages. Mapping translation has the potential to build bridges among the past (but still used) and new solutions to improve interoperability.

## Acknowledgments

The work presented in this paper is partially funded by Knowledge Spaces project (Grant PID2020-118274RB-I00 funded by MCIN/AEI/ 10.13039/501100011033); and partially funded by the European Union's Horizon 2020 Research and Innovation Programme through the AURORAL project, Grant Agreement No. 101016854.

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
