# OpenReview forum: "Devising Mapping Interoperability with Mapping Translation"
_kg-construct.github.io/KGCW/2022/Workshop — KGCW 2022_

### Official Review · ~Dylan_Van_Assche1 · 2022-03-25
**Interesting position paper on Mapping Interoperability**

**Rating:** 6
**Confidence:** 4

**Review:**

This position paper discusses the possible approaches for translating mapping languages to achieve interoperability among the various existing mapping languages such as RML, xR2RML, SPARQL-Generate, ShExML, etc. The authors properly introduce the State of the Art and develop the concepts of mapping translation which are proposed by Corcho et al.

I really enjoyed reading this paper and I clearly understood the possible mapping translation approaches & techniques.

# Major comments

I don't have a lot major comments except one: I missed the 'so what' of the paper.
I think the motivation needs to be better explained in the abstract and introduction.
If the authors would include here the benefits of mapping interoperability for the reader, the paper would improve significantly.

# Minor comments

- p1: 'that consist in translating' --> consists of
- p2: 'Helio mappingsHelio' --> seems to be a typo
- p2: 'Finally, Section 4 draw' --> draws
- p4: R2RML and OBDA mappings of Ontop are bidirectional translatable, it reads currently only as R2RML --> Ontop OBDA mappings. Their mappings are not really proprietary, I would not use that word here because the Ontop source code is fully open so you can see how these mappings are used etc. Commercial tools such as Stardog have also mappings, but they are completely closed source.
- p4: The enumeration (2) does not start with a capital like (1) and (3). I would make this consistent such as no capitals here.

---

### Official Review · ~Umutcan_Simsek1 · 2022-03-31
**position paper without a clear position**

**Rating:** 6
**Confidence:** 4

**Review:**

The paper points out the issue of mapping language heterogeneity and its impact on interoperability between knowledge graph construction pipelines. They then discuss different approaches for aligning different mapping languages as well as their trade-offs.

Although the paper does not have any groundbreaking observations, still makes valid points. The categorization and discussion of different approaches look technically sound. Therefore I am inclining towards the positive side of borderline. One thing that prevents me to give a higher score is that the position paper does not have a clear position. The authors discuss different approaches and propose different implementations for mapping translation, however I do not see the position of the authors clearly stated anywhere. Which one of these approaches the authors most likely go forward in their future endeavors, under which conditions? I think this question should be answered more clearly, if the paper is accepted.

Here are some rather specific comments:
- In the introduction, the authors claim that they propose "different techniques to extend mapping language translation to different languages". I cannot tell which section is doing this. Section 2 represents high-level approaches for mapping translation and Section 3 presents different ways to implement those. I do not see where the mapping translation techniques are "extended".
- Is the content of Section 2 based on the referenced paper from Corcho et al? It is not really concretely specified what is proposed in that paper as the concept of "mapping translation".
- In Section 3, there is a statement such as "since many languages are RDF-based, that is, follow the schema of an ontology...". May be a bit nitpicking, the being RDF-based does not necessarily imply that the data follow the schema of a specific ontology.  It just implies that data is in triple format with specific restrictions on what can go into the each element of the triples (to put in a very simplistic manner).
- In the conclusion section, the authors mention that the paper updates the scenarios mentioned in the referenced paper. Hard to say how this is done without reading that paper, which harms the self-contained nature a paper supposed to have (also related to my second comment.).
- There is a typo on page 4: such is the case of Helio-> such as the case of Helio

---

### Official Review · ~Miel_Vander_Sande2 · 2022-03-31
**No contribution**

**Rating:** 3
**Confidence:** 5

**Review:**

This paper surveys the different approaches for mapping non-RDF data into RDF to indicate that the current landscape is diffused. The authors claim that mapping translation can help overcome this issue. Or do they? Because it is really hard to tell what the authors propose or what this paper contributes. The real questions remain unanswered: what is mapping translation exactly? How does it look like and what are the exact benefits? What are its shortcomings? (there are significant mismatches in expressivity between eg. RML and SPARQL-Generate).

The paper i poorly written, which I know is not easy, but in this form, I just can't make much sense of it. Many sentences are convoluted, unclear or simply incomprehensible, including:
- S1;p2: "Mapping languages represent the relationships between the structure or the model of heterogeneous data and an RDF version following an ontology, i.e., the rules on how to translate"
- S1;p2: "an addressed specific feature"
- S2;p4: "This position paper develops the concept of mapping translation, proposed by Corcho et al. [16], a concept that can enhance the interoperability among existing mapping languages and thus, improve the user experience of these technologies by allowing communication and understanding among them."
- S3;p3: "However, it would leave out languages with other schemas, such as ShExML andSPARQL-based, wthout relying on software-based solutions."
Some terms seem poorly chosen:
- peer-to-peer translation: do you mean language-to-language or shorter, direct translation?


I was also wondering what the exact relation to Corcho et al. [16] is? What does this paper add on top of this paper? The different languages are piled up: transpiled domain-specific languages like YARRRML and SHexML (which are IMO a totally different beast), data-driven languages like R2RML, result-driven languages like SPARQL-Generate, but the authors don't offer any concrete paths to take for any of these combinations. Hence, the paper would be better of as a short survey paper about mapping languages and existing translation methods, if any.

Some other comments:
- the two first paragraphs in the intro should be connected better
- WoT should be properly explained
- when the authors state that the reader should be faniliar with current mapping languages, which ones do they mean?
- what's the difference between 'point-to-point' and 'family of languages'?
- one of the major benefits of using SPARQL as a translation/mapping language, is that you can also do value cleaning or other low-level transformations. This is not mentioned

---

### Official Review · ~Aidan_Hogan1 · 2022-04-01
**Position paper on an interesting and relevant topic; maybe lacks details (even for a position paper)**

**Rating:** 7
**Confidence:** 4

**Review:**

# Summary

This position paper discusses challenges relating to mapping translations, i.e., translations between mapping languages themselves. A wide range of mapping languages (for RDF) have been proposed down through the years, but they support different features, target different scenarios, are supported by different tools, etc. The question then is: how can we make mapping interoperable? The paper first introduces a number of different mapping methodologies, languages and scenarios. It then turns to mapping translation, and presents three different high-level techniques for such translation: manual procedural (e.g., Python scripts), manual declarative (e.g., SPARQL construct), and automatic (e.g., using ontology mapping). The conclusions then discuss the different approaches, and highlight the challenges relating to mapping translations.

# Strengths

S1: Very relevant and timely topic for discussion at this workshop.

S2: The topic itself is new to me, and I agree it is an important problem to look at.

# Weaknesses

W1: As a position paper, I might have liked more discussion of concrete scenarios where mapping translation is required, or in what areas such translation could have the most impact, and for what reason. (Note: I agree with the claim that this is an important issue to address, but I felt the paper could have been stronger and more specific in its motivations.) I might have also liked more details on some concrete next steps to embark on. The ending of the paper feels a bit too "open" to really motivate people to work on the topic. What do you think people should be working on in this direction?

W2: I did not fully understand the distinction that the paper draws between "Construct query-based translation" and "Executable mapping-based translation". I understand that the executable mappings are automatic, the underlying process would still seem to be construct query-based. I think maybe this categorisation could be made clearer, e.g., to divide into manual (software/procedural and query/declarative) and automatic (e.g., ontology based) approaches. This would appear to be the key distinction?

# Observation

O1: Figure 1: what about chains of mappings? All that is necessary is for the graph to be strongly connected, and then we can translate from any format to any other format we want (potentially via various intermediate formats), right?

O2: My sense is that what is needed here is not another mapping language or syntax, but maybe a theoretical or more abstract framework for mapping languages that defines their common core. (A loose analogy would be the way RDF has many syntaxes, but it does not really matter as they all parse to the same RDF data model with sets of triples.) If the languages (or some languages) can be defined in terms of the same core, then at least within that core, they can be made interoperable.

# Assessment

Though the paper lacks concrete details in terms of motivation, possible ways forward, etc., it is a position paper, and one that I think could generate interesting discussion at the workshop. Hence I lean towards accepting.

# Minor comments

- "Virtualization, (Ontology Based Data Access)" I would suggest: "Virtualization, (for example, Ontology Based Data Access)" as not all virtualization requires an ontology. We could virtualize based on, for example, an (R2)RML mapping.
- "the wide corpus" -> "the wide range" I think
- "WoT-mappings [8]<, which were> designed"
- "allows <for> the"
- "Finally, Section 4 draw<s>"
- "but <dissimilar> among languages"
- "These translation<s>"
- "need <for> learning"
- "Mapeathor, <which>" ... "<and thus> aligns with"
- "automatically done<> from"
- "it does not present <>an easy task," or "it does not present <itself> as an easy task,"

---

### Decision · Program_Chairs · 2022-04-11

**Decision:**

Accept

**Comment:**

Dear authors,

Thank your for submitting your paper. We are happy to inform you that we accept your paper! Please carefully consider the reviews when you prepare your paper for the camera-ready version. You will receive specific instructions to submit your camera-ready soon.

Kind regards
Organizers of the Knowledge Graph Construction workshop 2022